# Ecotoxicity of Single-Use Plastics to Earthworms

**Teresa Rodríguez [1], Dana Represas [1] and Emilio V. Carral [2],***

1  Department Zoology, Genetics and Physical Anthropology, University of Santiago de Compostela, 27002 Lugo, Spain
2  Department Functional Biology, University of Santiago de Compostela, 27002 Lugo, Spain
*  Correspondence: emilio.carral@usc.gal

**Abstract:** The excessive use of plastics in recent years, especially so-called single-use plastics, has led to an incipient increase in the presence of this material in the soil. As soil is the essential production factor in agriculture, this study aims to test the toxicity to earthworms of different concentrations of plastics using the same substrate for each sample. Earthworms are the main bioindicator of soil quality and are of particular ecological value because their disappearance or loss in population would have disastrous consequences for the environment. This study examines the growth, mortality and reproductive cycle of individual earthworms. The species to be studied is *Eisenia fetida* and during the test, individuals are immersed in five different concentrations: 1 mg/kg; 10 mg/kg; 100 mg/kg; 1000 mg/kg; and 2000 mg/kg. Bioplastic OK industrial compost (that meet the requirements established to comply with compostability according to EN 13432) was used. Whether the plastic in any of those concentrations is harmful to the worm population was then be checked. The test shows the mixture of PLA and PBAT (biodegradable plastic) studied has no significant effect on the population of *Eisenia fetida*. None of the variables studied yields significant data on this plastic and the effect it causes on the population of earthworms, *Eisenia fetida*.

**Keywords:** plastic; soil; survival; concentration; growth; earthworms

## 1. Introduction

One of the most important problems facing modern societies is the treatment and disposal of excessive waste generation. In this sense, the European Waste Framework Directive (Directive 2008/98/EC, 2008) [1], covers several activities, including prevention and preparation for the reuse and recycling of waste, to reduce the amount of waste produced and to carry out a waste management policy within the community (Domínguez and Edwards 2011) [2].

Most human activities now are based on or influenced by plastics (Souza et al., 2020) [3], such that, in 2018, about 360 million tons were produced. Because plastics are low-cost and often single-use, plastic pollution has become a serious environmental problem (Horton et al., 2017) [4] and has reached the planet's different ecosystems in a variety of ways. In the edaphic environment, a primary entry route is using organic fertilizers. The presence of plastics in the natural environment is rapidly increasing and represents a threat to our ecosystems so understanding their impact is essential to avoid potential environmental pollution (Censi, 2022, González-Pleiter, 2019) [5,6]. The edaphic ecosystem is not spared from this contamination and although it is considered to be a major transport pathway for microplastics (Zhang et al., 2019) [7], biotic toxicological testing and disposal in soil were not considered a priority area of research until recent years. Agricultural application of plastic products and widespread disposal by the general population lead to the occurrence of plastic residues in soil, so studies on the effect of bioplastics in the terrestrial environment (Mai, 2018; Quin, 2021; Zhang et al. 2018; Zhou et al., 2018) [8–11], as well as research on the risk of their transfer to the human food chain are increasing.

The edaphic environment constitutes a fundamental ecosystem for life because structures are settled in it, and we obtain from it the food we need to survive. It is also the habitat of a multitude of species that play a key role in its functioning by participating in the nutrient cycles, water cycle and activity of plant species, through favourable rooting, and other symbiotic behaviours (Casquero, 2017) [12]. A part of this edaphic biota is formed by earthworms that play an important role in fertility, edaphic structure, and maintenance of soil functions (OECD, 2004) [13]. These invertebrates have, in general, high reproductive rates and great adaptability, but are vulnerable to pollutants and can bioaccumulate edaphic pollutants, so they are widely used in edaphic toxicity studies (Jiang et al., 2020) [14]. To this must be added that they can be affected by the presence of these plastics in the soil, either through toxic effects on their bodies or by altering the physicochemical properties of the soil, which would affect their habitats. Not only earthworms, but all soil organisms exposed to microplastics can suffer lethal and sublethal effects such as damage to the intestinal tissue, lower growth rate or lower reproductive success (Holzinger et al., 2022) [15].

In recent years, because of pressure from consumers and legislation in different countries, the use of environmentally friendly packaging has become widespread. To improve companies' image, therefore, many organizations are developing and using new materials such as biodegradable and compostable polymers through industrial or domestic composting (OK Compost accreditation), which are used to produce all types of quick packaging such as plates, forks or cups or long-life items such as bottles, capsules and trays for food. In this regard, in 2018 the European Union published the document "A European strategy for plastics in a circular economy," which states that in 2030 the demand for recycled plastic will have to quadruple that of 2015 and that 100% of plastic packaging in the community will have to be reusable or recyclable (mechanical or organic recycling if compostable are included). In Spain, the use of non-compostable lightweight plastic bags has been banned since 1 January 2020 (RD 293/2018) and banning the marketing of single-use utensils (plates, cups, forks etc.) that are neither compostable nor biodegradable is already approved or in the approval process.

There are different types of plastics, such as bioplastics, which, as indicated in the literature, refer to materials produced from vegetable fats and oils, recycled food waste, etc., which can be divided into biodegradable, bio-based, or both. Biodegradable bioplastics are degraded in aqueous fluids under the effects of bacterial activity (Censi 2022) [5]. These include PET (Polyethylene terephthalate) PTT (polytrimethylene terephthalate) (from a natural renewable source), PBAT (polybutylene adipate terephthalate, which is biodegradable, and it decomposes in a short time) or PLA (polylactic acid), PHA (polyhydroxyalkanoates) or PBS (Polybutylene succinate), which are of natural renewable origin and biodegradable. In general, most biodegradable plastics are also bioplastics.

The properties of bioplastics have allowed for their use in many industrial sectors such as food packaging, but the significant environmental impact is a major problem at the end of their useful life if they are not properly managed. We are faced with products that make life easier for society but can have adverse effects on the health of living beings when they appear in nature, even in low concentrations (Careghini et al., 2015) [16]. Moreover, attention must be paid to the additives used in these biodegradable plastics to ensure they do not pose a danger to the natural environment, given that some biodegradable plastics degrade well in compost, but not in soil and vice versa (Flury and Narayan 2021) [17]. It must also be considered that environmental and soil conditions vary considerably from one place to another (Haider et al., 2019) [18], so a specific study cannot be applied to every part of the planet. Therefore, it is important to study the effect of these products, bearing in mind the need for an effective solution. Education also is necessary to mitigate the risk of littering and promote the proper collection of biodegradable plastics after use. Contamination reactions can be monitored by studying the viability, weight loss or reproductive impact on some soil and/or aquatic organisms. In this study, we worked with PLA, PBAT, TPC (thermoplastic copolyester made from 50% renewable sources based on rapeseed oil) and PHAs (polyhydroxyalkanoates) and earthworms (*Eisenia fetida*). The

assay was designed to check the environmental impact of bioplastic using an earthworm's lifecycle as a bioindicator.

PLA is a thermoplastic biopolymer whose precursor molecule is lactic acid. It has the greatest potential to replace conventional plastics because of its excellent physical and mechanical properties and because it can be processed using existing machinery with only minor adjustments (Munilla and Carracedo, 2005) [19]. It is a biodegradable plastic accredited for OK industrial composting and anaerobic digestion. PHAs is a biopolymer accredited for OK composting (industrial and home) and soil digestion. TPC is highly resistant to chemicals and ageing, and PBAT is a biodegradable copolyester that is an ideal blend as a component of bioplastics.

## 2. Materials and Methods

The OECD 207 standard (Short-term toxicity in terrestrial invertebrates) (OECD, 1984) [20] was followed for this study, according to which adult earthworms are exposed to different concentrations of the test material mixed with a substrate defined by the standard itself. This artificial soil was used to exclude the possibility that the soil may contain plastic particles, cocoons or earthworms and is fully accredited to evaluate the biotoxicity of pollutants. The substrate was prepared by the company Ecocelta. The species chosen for the test was *Eisenia fetida*, one of the species proposed by the European Organization for Cooperation and Development (OECD, 1984) [20] and by the International Organization for Standardization (ISO, 2013) [21] to evaluate the toxicity of all types of substances in the edaphic environment. The specimens were acquired from the company Ecocelta and were subjected to an acclimatization process using the substrate and environmental conditions in which the study was to be carried out. All earthworms were kept in culture chambers in the dark under controlled temperature conditions of $21 \pm 0.5$ °C. All the laboratory work was carried out in the Biological Specialties Support Section of the Research Infrastructures Area of the University of Santiago de Compostela at the Lugo Campus (Galiza, NW Spain). The tests for the accreditation OK industrial compost of the bioplastics used in this study were carried out by Instituto Tecnológico del Plástico (https://www.aimplas.es/ (accessed on 20 November 2021)), and this institute checked the biodegradability and compostable conditions of bioplastics following UNE-EN ISO 17556 and UNE-EN ISO 14855-1 standardized test. As shown in Table 1, they are biodegradable (they decompose under the enzymatic activity of microorganisms together with physical agents in environmental conditions that occur in nature) and compostable (they are transformed into compost through a composting process in a controlled time and under certain conditions).

**Table 1.** Composition of the samples used in this trial. PLA (polylactic acid), PBAT (adipic acid copolyester), TPC (thermoplastic copolyester made from 50% renewable sources based on rapeseed oil) and PHAs (polyhydroxyalkanoates).

| Code | Formulation | Biodegradation $\geq$ 90% (365 Days) | Disintegration $\geq$ 90% (180 Days) | Ecotoxicity |
|------|-------------|--------------------------------------|--------------------------------------|-------------|
| M1 | 60% (PLA + PBAT) + 40% PLA | OK | OK | OK |
| M2 | 80% (PLA + PBAT) + 20% PLA | OK | OK | OK |
| M3 | 50% PLA + 50% PHA | OK | NO | OK |
| M4 | 70% (80%PHA + 20%TPC) + 30% PLA | Ok | OK | OK |
| M5 | 60% PLA + 40% PHA | OK | OK | OK |
| M6 | 80% (PLA + PBAT) + 20% TPC | OK | OK | OK |

Once the biodegradability and compostability were accredited, we proceeded to the ecotoxicity study (compost quality) on earthworms. This test evaluates the exposure to pollutants and measures the biological effects resulting from such exposures in terms of

mortality, reproduction, growth and behavioural variations (de Andrea, 2010 [22]). The ecotoxicity study was carried out in two ways: contact toxicity test and soil toxicity test.

### 2.1. Contact Toxicity Test

The contact toxicity screening test identifies substances potentially toxic to earthworms and should be performed in conjunction with the artificial soil toxicity test.

The purpose of this test is to observe the effect of different concentrations of each of the plastics when they come in contact with earthworms through the epidermis. It followed the OECD guidelines and was carried out at three concentrations C1, C2, C3 (0.0007, 0.007 and 0.07 g/mL, respectively) with the addition of two controls (ZERO, BLK). Seven replicates per concentration and plastic were performed.

The tests were carried out in Petri dishes in which two circular filter papers of the same diameter as the plates were inserted. Once the 3 solutions of each of the plastics were prepared in ethanol, at different concentrations, 1 mL of the solution, ethanol in the case of the control, was pipetted onto the filter papers, making sure they were completely wet. The ethanol was allowed to evaporate, 2 mL of distilled water was pipetted onto all the plates, and an individual worm, clean and dry, was introduced between the two pieces of paper. The plates were placed in culture chambers at 21 °C for 24, 48 and 72 h, respectively, and the survival of the worms was checked and noted. They were considered dead when they did not respond to gentle mechanical stimuli on the anterior part.

### 2.2. Soil Toxicity Test

For this test, a procedure based on the OECD methodology was followed (OECD, 1984) [20]. Microcosms were created using 500 mL glass jars. Tests were performed at 5 different concentrations (1, 10, 100, 1000 and 2000 mg of each of the plastics per kg of soil), plus two controls (one of soil—ZERO—and a blank—BLK—supplied by the company). Three replicates per concentration were prepared with 400 g of soil to which the corresponding proportion of plastic was added. The media were prepared 48 h before inoculating the worms to avoid the possible increase in temperature caused by the activation of the bacterial flora. After this time, 7 clitellate earthworms were introduced into each of the microcosms, which were kept in dark conditions at $21 \pm 0.5$ °C. After 7 and 14 days, each microcosm was checked to verify the status of the specimens (% survival and % mortality).

### 2.3. Population Dynamics of Eisenia fetida

In ZERO, BLK and one of the plastics chosen at random, we monitored the evolution of worm weights and worm reproduction (number of cocoons and juveniles) for 84 days. The percentage increase in the number of total individuals in all samples was also calculated using the formula

$$\% \text{ INCREASE INDIVIDUALS} = (I_{84} - I_0) \times 100/I_0$$

where $I_{84}$ represents the number of individuals at the end of 84 days (end of the study) and $I_0$ the number of individuals at the beginning of the study. Calculations were also made to find the increase in the number of adults between the beginning and the end of the trial, as well as their weights according to the following formulas:

$$\% \text{ INCREASE NUMBERS ADULTS} = (A_{84} - A_0) \times 100/A_0$$

where $A_{84}$ and $A_0$ represent the number of adults at the end and the beginning of the trial.

$$\% \text{ INCREASE WEIGHT} = (P_{84} - P_0) \times 100/P_0$$

$P_{84}$ y $P_0$ represents the earworm adult weights at the end and the beginning of the bioassay, respectively.

Finally, the percentage of total adult growth was calculated for each of the cases, according to the following formula, which relates the weight and number of individuals according to the formula:

% GROWTH TOTAL = (final weight − initial weight) × 100/(final weight × nº individuals)

During the entire duration of the trial (84 days), the worms were not fed.

### 2.4. Statistical Analysis

The level of statistical significance was established at $p = 0.05$. The normality and homogeneity of variance of the data series were previously checked. If they were normal and homogeneous, ANOVA (MSD) was performed. If they did not present homogeneity of variance, the Games–Howell test was applied. If the data did not follow a normal distribution and did not present homogeneous variance, the Kruskal–Wallis nonparametric test was used. The SPSSv.2.5 statistical package was used (IBM).

## 3. Results

### 3.1. Contact Toxicity Test

As explained in the Section 2, a leakage test was performed with the CERO substrate, BLK and the M1–M6 plastics.

No incidence was observed because there was no leakage of individuals. The worms also responded to stimuli during the tests, but in some cases (M1_C1 and BLK_C1) a decrease in size was observed, as well as a low response to the stimuli. It should be noted that M1 is the bioplastic with the highest PLA content, according to data provided by Aimplas.

These results do not agree with those of other authors, such as (Ding et al., 2021) [23], who point out that earthworm evasion rates increased sharply with microplastic concentration, and that evasion behaviour was lower in PLA than in other microplastics.

### 3.2. Soil Toxicity Test

In this section, the survival of the worms at 7 and 14 days was studied, as indicated in the OECD 207 standard. The trials in which mortality occurred are shown in Table 2; the deaths occurred in the second week after the start of the study and in the highest concentrations (C4 and C5) except in the samples M1_C1_14 days and M5_C1_14 (lowest concentrations) and M2_C5_7 and M4_C5_7 (in the 7-day control). The occurrence of most of the losses in the 14-day control may indicate less survival of the worms when the length of exposure increases and the concentration of plastic in the substrate is higher.

The time of exposure to plastics (7–14 days) did not significantly influence worm mortality. The concentrations in which some were low were the highest (C4 and C5), reflecting the extreme concentrations usually found only in urban soils contaminated by waste (Ding et al., 2021) [23]. According to some authors (Zhang et al., 2018b) [24] worms could ingest higher proportions of biodegradable microplastics such as PLA because of their higher microbial load, high degradability, and higher nutritional content, which perhaps reduces their toxicity.

As we have described, the effect of different plastics on the worms' weight and survival was studied. From the results obtained and the absence of significant differences, we can say that it does not significantly affect survival, which coincides with previous studies (Domínguez et al., 2016; Verdú et al., 2018) [25,26] that concluded that this parameter is not very sensitive to ecotoxicology studies, although it should be emphasized that the toxicity of any substance in the soil is closely related to environmental variables and the physicochemical properties of the soil, especially clay and organic matter. It would therefore be interesting to add organic matter to the "pollutant" to evaluate its real toxicity (Verdú et al., 2018) [26], which might suggest some mechanism of self-regulation and physiological protection by the earthworms (Ding et al., 2021) [23].

**Table 2.** Samples and concentrations in which there was mortality in the soil test (M1–M6 samples, C1–C5: concentrations; R1–R3: replications; Total_indiv.: total individuals; R1–R3: replications).

|  |  |  | R1 | R2 | R | Total_indv. | % Survival | % Mortality |
|---|---|---|---|---|---|---|---|---|
| M1 | 14 days | C1 | 7 | 6 | 7 | 20 | 95.24 | 4.76 |
|  |  | C4 | 7 | 7 | 6 | 20 | 95.24 | 4.76 |
| M2 | 7 days | C5 | 7 | 7 | 6 | 20 | 95.24 | 4.76 |
|  | 14 days | C5 | 7 | 7 | 6 | 20 | 95.24 | 4.76 |
| M3 | 14 days | C4 | 6 | 7 | 7 | 20 | 95.24 | 4.76 |
|  |  | C5 | 7 | 7 | 6 | 20 | 95.24 | 4.76 |
| M4 | 7 days | C5 | 7 | 7 | 6 | 20 | 95.24 | 4.76 |
|  | 14 days | C4 | 7 | 7 | 6 | 20 | 95.24 | 4.76 |
|  |  | C5 | 7 | 7 | 6 | 20 | 95.24 | 4.76 |
| M5 | 7 days | C1 | 6 | 7 | 7 | 20 | 95.24 | 4.76 |
|  | 14 days | C1 | 6 | 7 | 7 | 20 | 95.24 | 4.76 |
| M6 | 14 days | C5 | 7 | 7 | 6 | 20 | 95.24 | 4.76 |

### 3.3. Population Dynamics of Eisenia fetida

For the population, dynamic study, M2 plastic, ZERO and BLK substrate were chosen. The test lasted three months, and the earthworm's weight, number of cocoons and juvenile individuals were checked.

First, the variation of weights in ZERO and BLK was studied. Figure 1 shows the data obtained. The starting weights were 7.43 g and 6.63 g, respectively, and as time passed, in the case of BLK, the weight increases only occurred until day 28, and from then on there was a continuous decrease until the end of the trial. The evolution of weight in ZERO increased until the weighing at 56 days, after which the weight began to decrease, perhaps from the reproductive effort made during the previous days, given that both copulation and laying involve large energy expenditure manifested by a weight decrease indicating an inverse relationship between weight and cocoon-laying (Graña 2016) [27]. It should be remembered that during this period, the specimens did not receive supplementary feeding.

Figure 2 shows the evolution of cocoon formation and the birth of new individuals; it can be seen that the population dynamics are better in CERO than in BLK.

To check for significant differences ($p = 0.05$) in the final weights between ZERO and BLK, an Analysis of Variance was performed, which did not release significant results.

As in the case of weights, an ANOVA was performed to check for significant differences in the number of cocoons and the number of juveniles ($p = 0.05$). No statistical differences were found.

For the M2 plastic, and for each of the concentrations, the same studies were carried out. It found the average weight of the adults increased in each of the concentrations up to 42 days and then began to decrease.

The evolution in the laying of cocoons that was not found until 28 days can be seen in Figure 3. In the low concentrations (C1–C3), the highest number was counted at 56 days, while in the highest concentrations (C4–C5) it was at 42 days, which may indicate the worms' adaptation to the higher concentrations of plastic was slower.

The first juveniles were observed at 28 days, which shows that cocoon-laying began after the 14-day check. At C1 and C5 concentrations, there was more consistent hatching, whereas there was more variation in the intermediate concentrations.

In general, worm hatching increased with exposure time, which is expected given their population dynamic. To check for significant differences ($p = 0.05$) between the different concentrations concerning the evolution of weights and population dynamics, an ANOVA was performed, and no significant differences were observed.

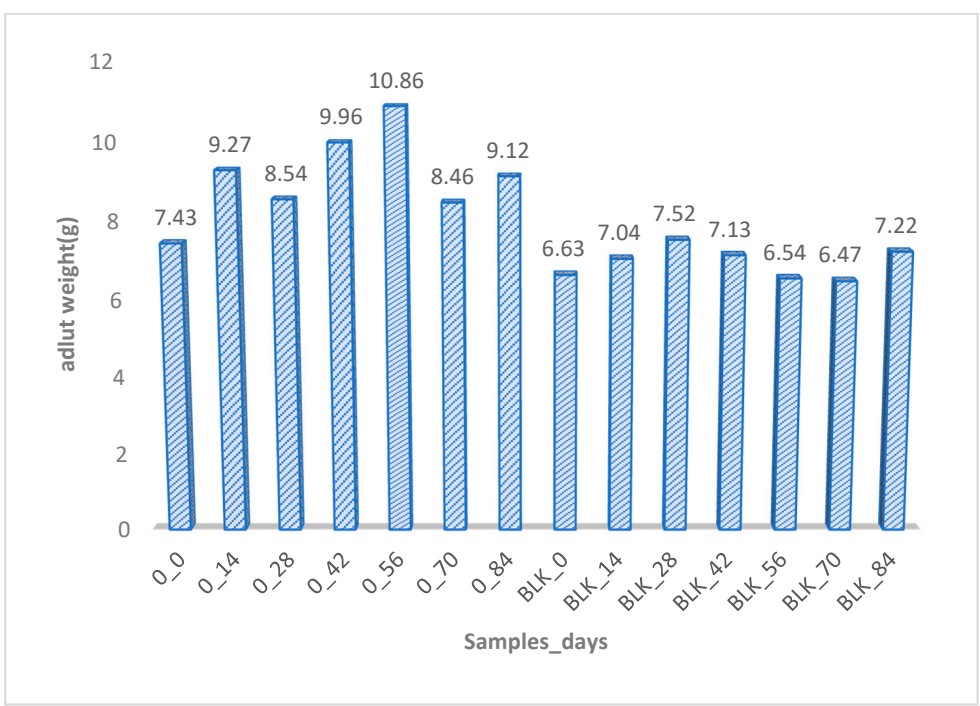

**Figure 1.** Evolution of weight, in grams, of adults during the duration of the trial in ZERO (0) and BLK.

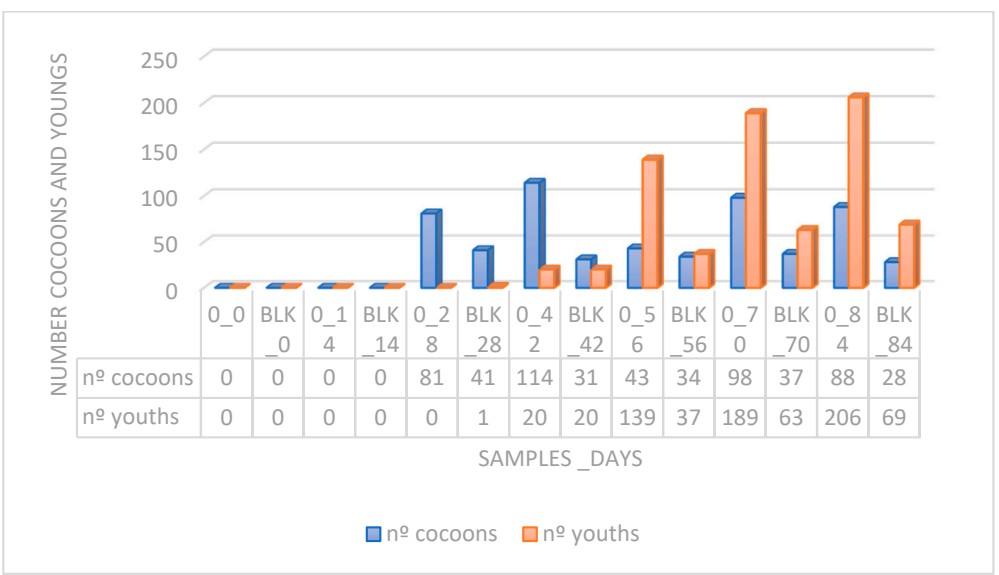

**Figure 2.** Evolution of the number of cocoons and juveniles in ZERO (0) and BLK from the beginning (0_0 and BLK_0) to the end of the trial (0_84, BLK_84).

Regarding the number of juveniles, statistically significant differences ($p \leq 0.05$) were estimated for the data as follows: ZERO showed positive significant differences for BLK and C1, C2 and C5 but not with C3 and C4, which seem to be the optimal levels for *E. fetida* development. Conversely, BLK showed negative significant differences with all concentrations (C1–C5) of M2, which means that population dynamics were worse in BLK than in the rest of the assays.

Some authors found significantly fewer cocoons and juveniles at low polystyrene concentrations, but the number remained similar to the control at high concentrations. This could have been caused by a higher weight at the beginning of the experiment and thus an

advance in the onset of reproductive activity. This does not agree with the results obtained in this study, given that the initial weights of C1 and C2 were like those of the other trials and in some cases even higher. The highest increase in the number of individuals always appears in ZERO (the total increase in the number of individuals was 1019.05%) and is the lowest in BLK (347.62%), with different concentrations in between. It should be noted that C3 (923.81%) and C4 (828.57%) show higher increases than C1 (64.86%), C2 (609.52%) and C5 (623.81%).

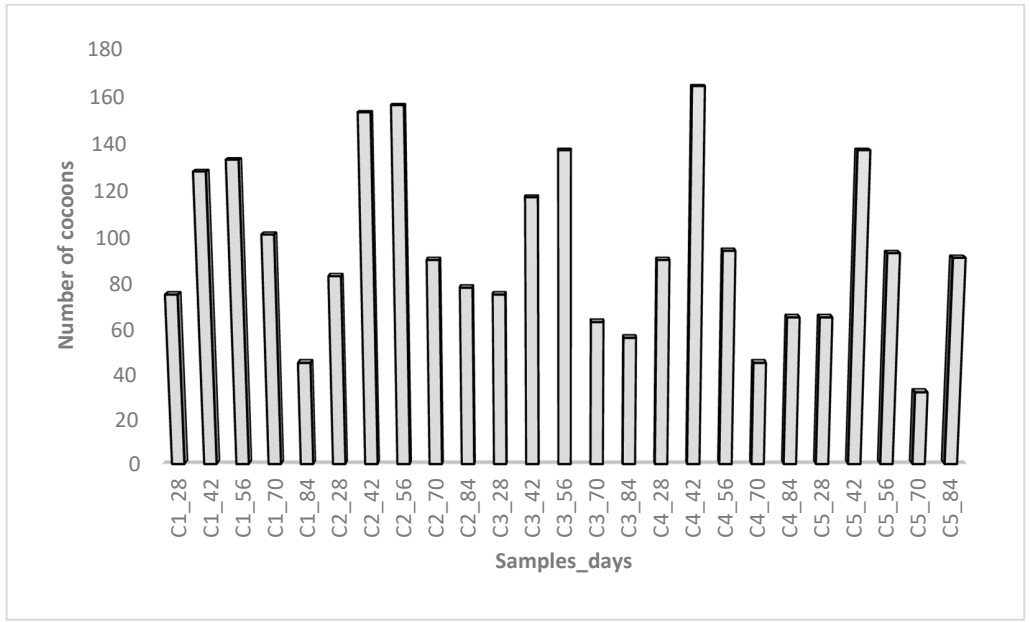

**Figure 3.** Evolution of cocoon-laying at the different concentrations (C1–C5) studied.

Subsequently, the percentage increase in the total number of individuals throughout the study was calculated, according to the formula provided previously to check for significant differences ($p \leq 0.05$) between the different samples. The results are shown in Table 3 and coincide, in most of the comparisons, with those obtained for the evolution of the number of juveniles.

**Table 3.** Presence or absence of a significant percentage increase in the total number of individuals.

|       | CERO | BLK | C1  | C2  | C3 | C4 | C5 |
|-------|------|-----|-----|-----|----|----|----|
| CERO  | –    |     |     |     |    |    |    |
| BLK   | YES  | —-  |     |     |    |    |    |
| C1    | YES  | YES | –   |     |    |    |    |
| C2    | YES  | YES | NO  | —   |    |    |    |
| C3    | NO   | YES | YES | YES | —— |    |    |
| C4    | NO   | YES | NO  | NO  | NO | —  |    |
| C5    | NO   | NO  | NO  | NO  | NO | NO | —  |

The increase in the number of adults and their weight between the beginning and the end of the trial was calculated and was positive increase in all treatments except C1 and C2.

Finally, the percentage of total adult growth was calculated according to the formula is provided in the Section 2 for all the trials and no significant differences between the different values were obtained; the lowest concentrations (C1, C2) show the lowest increase, even negative.

In general, this study did not show any unusual earthworm behaviour for the chosen bioplastic (PLA base). Earthworms appear to seek out small biodegradable microplastics for feeding, although it is unclear whether they harm the oligochaete gut and consequently their effect on the weight or population dynamics of oligochaetes (Zhang et al., 2018b) [24]. In addition to all this, microplastic particles are a potential risk for the survival of oligochaetes because they can accumulate in the digestive tract and affect feeding behaviour and even development (Rodríguez-Seijo et al., 2017) [28]. It is also unknown what kind of decomposition the biopolymers undergo both in the digestive tract of the worms and in their galleries (Huerta-Lwanga et al., 2021) [29].

Several authors have pointed out that exposure to microplastics has a negligible effect on female reproductive organs but in the male may affect spermatogenesis and sperm density, which may be the case for earthworm reproduction (Kwak and An, 2021) [30]. There are studies reporting growth inhibition in *E. fetida* after long-term exposure to polypropylene microplastics (Zhou et al., 2020) [31].

Earthworm mortality, reproduction and growth depend on the concentration of plastic particles in the soil, but at concentrations up to 1000 mg/kg, no effects on mortality and reproduction have been recorded (Liwarska-Bizukojc, 2021) [32].

Earthworms exposed to microplastics lose weight, but no relationship between this loss with microplastic concentration has been established, nor have significant differences been demonstrated in the growth rate of *E. fetida* when soil concentrations of microplastics are raised to 1, 5, 10 and 20% *w/w* compared with the control (Wang et al., 2019) [33].

On the other hand, the presence of PLA seems to influence the distribution of earthworms vertically in the soil, as they migrate deeper into the soil, which is unusual for epigean species and could represent a problem for the structure and functioning of the soil ecosystem (Liwarska-Bizukojc, 2022) [34]. The transport of bioplastic particles by earthworms, however, can affect other contaminants present in the soil (Liwarska-Bizukojc, 2021) [32]. Microplastics also accumulate and are transferred in the food chain. Concentrations of 100 or 1000 mg/kg dry weight of soil (corresponding to our C3 and C4) influence the accumulation of plastics in earthworms (Jiang et al., 2020) [14]. It is clear from all the studies conducted that the biosafety of biodegradable plastic film must be verified (Sintim et al., 2019; Zhang et al., 2019) [7,35].

## 4. Conclusions

In general, in this study, the earthworms did not show unusual behaviour with the chosen bioplastic (PLA base). Mortality was not a parameter particularly sensitive to contamination by PLA-based bioplastics, both when individuals were exposed indirectly (soil toxicity test) and directly (contact toxicity test). Growth and reproduction were less affected at intermediate tested concentrations (C3, C4)

However, this trial raises the need for such studies in more than one species of earthworm or using other soil bioindicators because the data obtained for a single toxin risk underestimating the impact of toxic substances on the natural environment.

This study and other similar previous studies provide information on the effect of a given product, but studies of contamination in soils should be carried out based on multiple contaminants to obtain realistic information on the danger given that soil organisms are exposed to more than one toxicant, and the interaction between all of them, together with environmental factors, can be very complex. It is, therefore, necessary to investigate the effects of the contaminants on the soil properties, and the biotic interactions that occur, to understand the risks to which the soil biota is exposed.

Our results come from laboratory assays, and show no toxicity to earthworms used as bioindicators, but toxicity level evaluation in field conditions is mandatory for finding the real environmental impact of bioplastic decomposition.

**Author Contributions:** Conceptualization, T.R. and E.V.C.; methodology, T.R. and E.V.C.; software, E.V.C. visualization, T.R. and E.V.C.; validation, T.R. and E.V.C.; supervision, T.R. and E.V.C.; data curation, D.R. and T.R. writing-original draft, D.R.; formal analysis, D.R. and E.V.C.; investigation, D.R., T.R. and E.V.C.; resources, T.R.; writing-review-Editing, E.V.C. and T.R.; project administra-

tion, T.R.; funding acquisition, T.R. All authors have read and agreed to the published version of the manuscript.

**Funding:** This research was funded by the project "Single-use plastic products with controlled biodegradability (BIO+)" in the framework of the CIEN CDTI 2017 program (2018-CE225).

**Data Availability Statement:** Data available on request due to restrictions (company privacy).

**Acknowledgments:** Our thanks to the Biological Specialties Support Section of the Research Infrastructures Area of the University de Santiago de Compostela at Lugo campus, and to its technician Mª del Pilar López Valín.

**Conflicts of Interest:** The authors declare no conflict of interest.

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
