# Peer review of "Ecotoxicity of Single-Use Plastics to Earthworms"

_environments, doi:10.3390/environments10030041_

Round 1

Reviewer 1 Report

Dear Editor

The investigation presented is very interesting, but I recommend these major revisions before accepting the manuscript

Line 16: Please, specify what is the substrate

Lines 38-39: I do not agree with the authors, they affirm that “there have been few studies on the effect of microplastics in the terrestrial environment”, but if they use a generic scientific search engine they find more than 3000 results in the past three years. If they want select the edaphic environment they must specify it better. I recommend to the authors revise this aspect.

Lines 69 – 77: I suggest to the authors a new recently published review that better clarifies the concept of biodegradability and bioplastics: it is not always true and simple the link between biodegradability and bioplastics.

Censi et al. - Bioplastics: A new analytical challenge - Frontiers in Chemistry 2022, 10, 971792

Lines 97 – 104: The authors affirm that biopolymers used are ok composting (industrial and home) but do not indicate any standard test methods and/or specific controlled environments showing these sentences. I also suggest better clarifying the biodegradability issue which is still a crucial and a very highly debated topic in the scientific community. Are the authors sure that the specific controlled environments used for determining the compost ability of polymers can guarantee the success of the process in the real condition of the edaphic ecosystem?

Lines 121 – 126: similar comments for lines 97 – 104. Please specify better

Lines 360 – 374. The authors conclude that this study is only a preliminary investigation of contamination in the soil, and I agree with them. So I suggest if it is possible to add a little comment at the end of paragraph 3, that can summarize all the results obtained, to make it easier to interpret the section itself.  

Author Response

Answer to Review 1

Thanks a lot for your very useful considerations that help to improve our manuscript. We think that all considerations are followed, and new paragraphs were incorporated into the manuscript in the sense of the review´s considerations.

Line 16: Please, specify what is the substrate

Bioplastics OK industrial compost

Lines 38-39: I do not agree with the authors, they affirm that “there have been few studies on the effect of microplastics in the terrestrial environment”, but if they use a generic scientific search engine they find more than 3000 results in the past three years. If they want select the edaphic environment they must specify it better. I recommend to the authors revise this aspect.

The presence of plastics in the natural environment is rapidly increasing and represents a threat to our ecosystems so understanding their impact is essential to avoid potential environmental pollution (Censi, 2022, González-Pleiter, 2019). The edaphic ecosystem is not spared from this contamination and although it is considered to be a major transport pathway for microplastics (Zhang et al., 2019), biotic toxicological testing and disposal in soil were not considered a priority area of research until recent years. Agricultural application of plastic products and widespread disposal by the general population lead to the occurrence of plastic residues in soil, so studies on the effect of bioplastics in the terrestrial environment (Mai, 2018; Qi, 2020; Quin, 2021; Zhang et al. 2018a; Zhou et al. 2018), as well as investigations on the risk of transfer of microplastics from terrestrial agriculture to the human food chain (Saker 2020), are increasing.

Lines 69 – 77: I suggest to the authors a new recently published review that better clarifies the concept of biodegradability and bioplastics: it is not always true and simple the link between biodegradability and bioplastics.

There are different types of plastics, such as bioplastics, which, as indicated in the literature, refer to materials produced from vegetable fats and oils, recycled food waste, etc.,  which can be divided into biodegradable, bio-based, or both. Biodegradable bioplastics are degraded in aqueous fluids under the effects of bacterial activity (Censi 2022)

Lines 97 – 104: The authors affirm that biopolymers used are ok composting (industrial and home) but do not indicate any standard test methods and/or specific controlled environments showing these sentences. I also suggest better clarifying the biodegradability issue which is still a crucial and a very highly debated topic in the scientific community. Are the authors sure that the specific controlled environments used for determining the compost ability of polymers can guarantee the success of the process in the real condition of the edaphic ecosystem?

Lines 121 – 126: similar comments for lines 97 – 104. Please specify better

The biodegradability and compostable conditions of bioplastics were checked by Instituto Tecnológico del Plástico following UNE-EN ISO 17556 and UNE-EN ISO 14855-1 standardized test (https://www.aimplas.es/).

Lines 360 – 374. The authors conclude that this study is only a preliminary investigation of contamination in the soil, and I agree with them. So I suggest if it is possible to add a little comment at the end of paragraph 3, that can summarize all the results obtained, to make it easier to interpret the section itself.  

Our results are coming from laboratory assays, and show no toxicity to earthworms used as bioindicators, but toxicity level evaluation in field conditions is mandatory to find the real environmental impact of the bioplastics decomposition.

Reviewer 2 Report

In general, the article is of importance and covers the Aims & scope of the Journal. However, in my opinion the article lacks some discussion in relation to existing reported advances and state-of-the art in respect to ecotoxicity of single-use plastics, which they shall be addressed in order to enhance the importance of this work.

Specific comments

·        English language shall be checked by a native speaker.

·        Why the capital is in Capital letters? Please follow Journal's guidelines.

·        In my opinion, significant publication on the field of biodegradation of plastics with insects are missing. Indicatively:

1.      Tsochatzis et al. Chemosphere 1, 2021. Biodegradation of expanded polystyrene by mealworm larvae under different feeding strategies evaluated by metabolic profiling using GC-TOF-MS. Chemosphere 281, 130840.

2.      Tsochatzis et al. 2022. Cellular lipids and protein alteration during biodegradation of expanded polystyrene by mealworm larvae under different feeding conditions. Chemosphere 300, 134420.

3.      Yang et al., 2015. Biodegradation and Mineralization of Polystyrene by Plastic-Eating Mealworms: Part 1. Chemical and Physical Characterization and Isotopic Tests. Env. Sci. Int., 49: 12080-12086.

4.      Yang et al., 2015. Biodegradation and Mineralization of Polystyrene by Plastic-Eating Mealworms: Part 2. Role of Gut Microorganisms. Env. Sci. Int, 49: 12087-12093.

5.      Yang et al. 2020. Biodegradation and mineralization of polystyrene by plastic-eating superworms Zophobas atratus. Sci. Total Env., 708: 135233.

6.      S.-S. Yang, W-M. Wu, A.M. Brandon, et al. 2018. Ubiquity of polystyrene digestion and biodegradation within yellow mealworms, larvae of Tenebrio molitor Linnaeus (Coleoptera: Tenebrionidae). Chemosphere, 212, 262-271.

7.      A.M. Brandon et al. 2018. Biodegradation of polyethylene and plastic mixtures in mealworms (larvae of Tenebrio molitor) and effects on the gut microbiome. Environ. Sci. Technol., 52 (2018), pp. 6526-6533.

8.   Lou et al.  2020. Biodegradation of polyethylene and polystyrene by greater wax moth larvae (Galleria mellonella L.) and the effect of Co-diet supplementation on the core gut microbiome. Environ. Sci. Technol., 54 (2020), pp. 2821-2831.

·    Experimental part: The environmental conditions for the growth of insects are important to be mentioned.

·        L.97-104: Please provide clear and concrete goals for this research project.

·        Tables 2 & 3 captions: Please place the caption above the Table and below. Please correct caption of Table 2 (not “Tabla”).

Author Response

 Answer to REVIEW 2

Thanks a lot for your very useful considerations that help to improve our manuscript. We think that all your considerations are followed, and new paragraphs were incorporated into the manuscript in the sense of the review´s considerations.

English language shall be checked by a native speaker.

Please, find attached the proofreading certification letter.

  • Why the capital is in Capital letters? Please follow Journal's guidelines.

Done

  • In my opinion, significant publication on the field of biodegradation of plastics with insects are missing. Indicatively:

Our assays were carried out following the ISO standards to use an earthworm (E. fetida ) as a bioindicator. We do not use insects. Then, we understand that the references in relationship with the earthworm are the relevant ones. In any way, new references were added.

  1. Tsochatzis et al. Chemosphere 1, 2021. Biodegradation of expanded polystyrene by mealworm larvae under different feeding strategies evaluated by metabolic profiling using GC-TOF-MS. Chemosphere 281, 130840.
  2. Tsochatzis et al. 2022. Cellular lipids and protein alteration during biodegradation of expanded polystyrene by mealworm larvae under different feeding conditions. Chemosphere 300, 134420.
  3. Yang et al., 2015. Biodegradation and Mineralization of Polystyrene by Plastic-Eating Mealworms: Part 1. Chemical and Physical Characterization and Isotopic Tests. Env. Sci. Int., 49: 12080-12086.
  4. Yang et al., 2015. Biodegradation and Mineralization of Polystyrene by Plastic-Eating Mealworms: Part 2. Role of Gut Microorganisms. Env. Sci. Int, 49: 12087-12093.
  5. Yang et al. 2020. Biodegradation and mineralization of polystyrene by plastic-eating superworms Zophobas atratus. Sci. Total Env., 708: 135233.
  6. S.-S. Yang, W-M. Wu, A.M. Brandon, et al. 2018. Ubiquity of polystyrene digestion and biodegradation within yellow mealworms, larvae of Tenebrio molitor Linnaeus (Coleoptera: Tenebrionidae). Chemosphere, 212, 262-271.
  7. A.M. Brandon et al. 2018. Biodegradation of polyethylene and plastic mixtures in mealworms (larvae of Tenebrio molitor) and effects on the gut microbiome. Environ. Sci. Technol., 52 (2018), pp. 6526-6533.
  8. Lou et al. 2020. Biodegradation of polyethylene and polystyrene by greater wax moth larvae (Galleria mellonella L.) and the effect of Co-diet supplementation on the core gut microbiome. Environ. Sci. Technol., 54 (2020), pp. 2821-2831.

  • Experimental part: The environmental conditions for the growth of insects are important to be mentioned.

All earthworms were kept in culture chambers in the dark under controlled temperature conditions of 21 ± 0.5 C

        L.97-104: Please provide clear and concrete goals for this research project.

The assay was designed to check the environmental impact of bioplastic using an earthworm life- cycle as a bioindicator.

  • Tables 2 & 3 captions: Please place the caption above the Table and below. Please correct caption of Table 2 (not “Tabla”).

Done

Round 2

Reviewer 1 Report

Just a few minor revisions

Abstract Section

 Line 16: the meaning of the following sentence is not understandable

 “ Bioplastic OK industrial compost was used”

 Revise the sentence and specify what type of bioplastics are used

 Reference Section

 Line 388 and Line 409: The typology of the bibliographic reference is not clear

 Please Revise

Line 393: The citation is not properly reported

Generally, it is necessary to standardize the method of citation of the whole bibliography

Author Response

Abstract Section

 Line 16: the meaning of the following sentence is not understandable

 “ Bioplastic OK industrial compost was used”

 Revise the sentence and specify what type of bioplastics are used

A new sentence was added to a better understanding of what “Bioplastic OK “ means.

 Reference Section

 Line 388 and Line 409: The typology of the bibliographic reference is not clear

 Please Revise

 Changes done.

Line 393: The citation is not properly reported

 Changes done.

Generally, it is necessary to standardize the method of citation of the whole bibliography

Bibliography revised

Reviewer 2 Report

In general the authors adequately answered some of my comments. Some minor comments to be considered:

- Abstract/L.16: What OK refers to? Please revise.

- The authors are studying the exotoxicity of single-use plastics. However, on L. 37-47, a newly introduced paragraph describes among others the "microplastics" transfer to the human food chain. This part needs revision. 

Author Response

Abstract/L.16: What OK refers to? Please revise.

A new sentence was added to a better understanding of what “Bioplastic OK“ means.

- The authors are studying the exotoxicity of single-use plastics. However, on L. 37-47, a newly introduced paragraph describes among others the "microplastics" transfer to the human food chain. This part needs revision. 

A new sentence was added to a better understanding of bioplastic research relevance.